# Clinical and Psychological Impact of COVID-19 Infection in Adult Patients with Eosinophilic Gastrointestinal Disorders during the SARS-CoV-2 Outbreak

**DOI:** 10.3390/jcm9062011

**Published:** 2020-06-26

**Authors:** Edoardo Vincenzo Savarino, Paola Iovino, Antonella Santonicola, Matteo Ghisa, Giorgio Laserra, Brigida Barberio, Daria Maniero, Greta Lorenzon, Carolina Ciacci, Vincenzo Savarino, Fabiana Zingone

**Affiliations:** 1Gastroenterology Unit, Department of Surgery, Oncology and Gastroenterology, University of Padua, 35128 Padua, Italy; matteoghisa@yahoo.it (M.G.); giorgiolaserra91@gmail.com (G.L.); brigida.barberio@gmail.com (B.B.); dariamaniero@gmail.com (D.M.); gretalorenzon90@gmail.com (G.L.); fabiana.zingone@unipd.it (F.Z.); 2Department of Medicine, Surgery, Dentistry, Scuola Medica Salernitana, University of Salerno, University Hospital San Giovanni di Dio e Ruggi d’Aragona, 84131 Salerno, Italy; piovino@unisa.it (P.I.); antonellasantonicola83@gmail.com (A.S.); cciacci@unisa.it (C.C.); 3Gastroenterology Unit, Department of Internal Medicine, University of Genoa, 16126 Genoa, Italy; vsavarin@unige.it

**Keywords:** EGID (Eosinophilic gastrointestinal disease), SARS-CoV-2, COVID-19, eosinophilic esophagitis, eosinophils, steroid, immunosuppression, coronavirus

## Abstract

Eosinophilic gastrointestinal diseases (EGIDs) are chronic gastrointestinal conditions requiring corticosteroid and immunosuppressive therapy for disease control. Patients with EGIDs usually report impaired quality of life. We aimed to report the clinical and psychological impact of COVID-19 infection in EGID patients. In this prospective web-based study we invited all consecutive EGID patients attending the University Hospital of Salerno (Campania) and Padua (Veneto) to fill an ad hoc COVID-19 survey. Moreover, a telemedicine service for direct consultation was organized. Data regarding the occurrence and perception of COVID-19 infection as well as clinical information were recorded. The study population included 102 EGID patients (mean age 36.6 years, 34 females), of whom 89 had eosinophilic esophagitis, nine had gastroenteritis, and four had colitis. No patient was diagnosed with COVID-19 or had recurrence of his/her primary disease. All of them were adherent to therapy and preventive measures adoption. Most patients were worried because of COVID-19 and social preventing measures but did not consider themselves at major risk or susceptible to COVID-19 or other infections due to their chronic condition or therapy. Female gender and low education level were associated to a higher psychological perception of COVID-19 compared to lockdown status or other demographic and clinical factors (*p* < 0.05). Overall, COVID-19 had a limited clinical impact on patients with EGIDs. The degree of education and sex, but not the fact of living in a lockdown area, influenced the perception of SARS-CoV-2 infection.

## 1. Introduction

Eosinophilic esophagitis (EoE) is a chronic immune/antigen-mediated disorder characterized by symptoms of esophageal dysfunction (i.e., dysphagia and bolus impaction) and, histologically, by eosinophilic inflammation in the absence of secondary causes of eosinophilia [1,2,3]. In the last two decades, EoE has transformed from a rare condition to a disease that is commonly encountered in clinical practice, representing a major cause of upper gastrointestinal morbidity [2]. The incidence of EoE has been investigated in several population-based studies, conducted primarily in North America and Europe, ranging from a low rate of 2.1/100,000/year in the Netherlands to a high rate of 12.8/100,000/year in Ohio in the United States [2]. The pathogenesis of EoE is partially understood and it seems that protein antigens, mainly of food derivation, stimulate a Th2-mediated immune response, able to recruit eosinophils, mast cells, basophils, and several inflammatory chemokines and cytokines [4,5]. Adult patients with EoE have substantially impaired quality of life due to on-going symptoms and treatments, including orally administered topical corticosteroids and proton pump inhibitors [6,7]. Moreover, in case of disease refractory to first-line medical therapy, systemic corticosteroids or biologic therapies are required, with the risk of increasing the rate of infection and complications [8]. Similar treatments are adopted for the other forms of eosinophilic gastrointestinal diseases (EGIDs), namely eosinophilic gastroenteritis (EGE) and eosinophilic colitis (EC). These last disorders are characterized by specific presentation depending on the location (organ) and extent (layer invasion) of eosinophilic infiltration (i.e., abdominal pain, nausea, vomiting, early satiety, and diarrhea) and have also been associated with impaired quality of life [9]. Compared to EoE, non-EoE EGIDs are quite rare, but the use of systemic glucocorticoids is quite common, making non-EoE EGID patients at higher risk of developing steroid-related adverse events.

The severe acute respiratory syndrome coronavirus 2 (SARS-CoV-2) responsible for coronavirus disease 2019 (COVID-19) was declared a pandemic in March 2020. After the outbreak in China, Italian Health Authorities reported the first two cases of person-to-person virus transmission in Codogno (Lodi, Lombardy) and Vo’ Euganeo (Padua, Veneto) on 21 February and 23 February 2020, respectively. Thereafter, northern Italy rapidly became one of the areas with the highest incidence of new cases, with more than 143,626 cases and 18,279 deceased as of April 9th [10]. Preliminary observations showed that children may have only a mild presentation of COVID-19 or are even able to recover while being asymptomatic, while the clinical course of COVID-19 disease in adults ranges from asymptomatic condition to severe disease with significant morbidity and mortality. In particular, mortality seems to be highly influenced not only by advanced age and male gender, but also by the concomitant presence of chronic diseases (i.e., diabetes, cardiovascular diseases, etc.) [11,12,13]. Recently, immune-mediated gastrointestinal diseases have been associated to SARS-CoV-2 infection and its complications, although the magnitude of this association is not huge and the significance is not univocal [14,15]. Indeed, some data highlighted that the increasing rate of mortality of patients with inflammatory bowel diseases is more related to ongoing systemic corticosteroid therapy rather than to the disease itself, leading various international gastrointestinal societies to recommend to tapering down or withdrawing systemic steroids, when possible [16].

Currently, no data are available on the clinical impact of COVID-19 infection on EGID patients. Similarly, no studies have been published to date aimed to understand the COVID-19 perception by patients with chronic gastrointestinal diseases, such as those with EGIDs. Indeed, the extraordinary measures applied to contain the viral spread, such as the stop of every activity and the lockdown of vast areas, together with the public attention and mass media information, may have influenced the physical as well as the psychological well-being of our patients [17].

With this premise, we aimed to investigate the clinical impact of COVID-19 infection in a sample of EGID patients and to assess the degree of psychological perception of the COVID-19 emergency from two cohorts of EGID patients, one living in a red-zone during the first weeks of lockdown and the other one in a low-rate infection area without the need of immediate lockdown.

## 2. Patients and Methods

In this prospective, observational, web-based study, between March 18th and April 4th we invited by e-mail EGID patients regularly followed at the Gastroenterology Units of the University of Salerno (Campania region, Salerno, Southern Italy) and the University of Padua (Veneto region, Padua Northern Italy) to complete an ad hoc web-based COVID-19 survey (Figure 1). In case of lack of response after the second e-mail recall, patients were directly contacted by phone calls or personal email/WhatsApp messages. All patients had provided their e-mail addresses and phone number in order to be contacted for filling questionnaires or receiving information regarding their diseases and needs during outpatient visits (EGID registry approved by the research ethics committee of the Azienda Ospedaliera di Padova under the protocol CESC 4204/AO/17). Moreover, all patients who accepted to be interviewed, signed informed consent for the collection, handling, and storage of data, which was included in the presentation of the questionnaire. Noteworthy, the Campania region was a low-prevalence COVID-19 infection area and was included in the national lockdown after about two weeks, whereas the Veneto region, which was immediately overwhelmed by the COVID-19 outbreak in northern Italy, was declared a red-zone area due to the high prevalence of this infection.

The web-survey included 14 multiple-choice questions aimed to evaluate the perception of COVID-19. In particular, we asked if patients were worried about the epidemic; if they felt more susceptible to the infection compared to the general population; if they preferred to avoid going to hospital or crowded places; if they felt disturbed or anxious because of coronavirus infection; and their thoughts about remote consultations. Moreover, we demanded if patients had undergone a flu vaccination in the previous months and if they desire to be vaccinated when a COVID-19 vaccine will be available. We asked if they or their family members or contacts had been suffering from coronavirus 19 infection or flu-like symptoms and had been diagnosed with COVID-19 based on nasopharyngeal swab. Finally, data about demographic and clinical features, including gender, age, time of EGID diagnosis, region of origin (Veneto or Campania), type of disease (eosinophilic esophagitis, eosinophilic gastroenteritis, and eosinophilic colitis), and symptom recurrence were recorded. We also asked for information on ongoing therapy (i.e., proton pump inhibitors, 6 food elimination diet, topical or systemic steroids, biologic drugs, immunosuppressive compounds, sodium cromoglycate, anti-leukotrienes) and co-morbidities at the time of web-survey.

The diagnosis of EoE was posed according to international criteria, i.e., presence of typical symptoms of esophageal dysfunction (dysphagia and/or food impaction) and at least 15 eosinophils per high-power field (eos/HPF) in biopsies taken from middle/proximal esophagus [18]. Likewise, the other forms of EGID were diagnosed according to international criteria [19]. Other causes of gastrointestinal eosinophilia were excluded (e.g., parasitic infection, gastro-esophageal reflux disease, drugs, rheumatic diseases, and malignancies) before final diagnosis. Also, patients younger than 18 years old were excluded from the analysis.

From the beginning of the outbreak our patients were strongly invited to follow the measures suggested by our National Health System and the National Association of EoE patients (ESEO) as well as the European Society of Eosinophilic Esophagitis (EUREOS), for the prevention of SARS-CoV-2 infection: handwashing, sanitization, avoiding public and overcrowded places, and wearing surgical masks in every public place. To note, during quarantine, the personnel of the two Gastroenterology Units advised patients to postpone the regular outpatient visits, if not urgent, and invited them to accept remote healthcare consultations, sending laboratory exams by e-mail in advance. Thus, health care providers scheduled phone calls to all patients, asking the status of well-being and discussing exam results. Similarly, endoscopic procedures were done only in patients suffering from moderate-to-severe symptoms. Moreover, according to our routine clinical practice, patients could communicate with our team through a dedicated e-mail or a phone number reaching a physician or nurse involved in EGID management.

All co-authors had access to the study data and reviewed and approved the final manuscript. The study protocol conforms to the ethical guidelines of the 1975 Declaration of Helsinki.

### Statistical Analysis

Categorical and continuous variables were expressed as proportions, percentages, and mean with standard deviation (SD), respectively. Comparisons among categorical and continuous variables were performed using the chi-square test and *t*-test, respectively. We tested for differences in survey’s answers within the following study population’s subgroups: gender, age at test, living area, time from diagnosis, type of eosinophilic diseases, presence of comorbidities, education, and current therapy. A *p*-value <0.05 was considered statistically significant. STATA 11 software (Stata Corp., College Station, TX, USA) was used for statistical analysis.

## 3. Results

A total of 25 and 132 adult patients with EGIDs attending the Gastroenterology Units of the University of Salerno (Campania region, Salerno, Southern Italy) and the University of Padua (Veneto region, Padua Northern Italy), respectively, who had provided the consent of being contacted, were invited to answer our web-based questionnaire. We received 102 answers out of 157 patients by e-mail (64.9%). More patients from Campania region completed the survey (80% vs. 62.1%), although a statistically significant difference was not reached when compared with those of the Veneto region (*p* = 0.1103). As illustrated in Table 1, the study population was composed by 20 patients from Campania (mean age 40.9 (14.1) years, 25.0% female) and 82 from Veneto (mean age 35.5 (11.8) years, 36.6% female), mostly from urban areas (71.0%), and with an academic degree (47.0%), without any difference between the two regions (*p* = ns). Moreover, most patients were younger than 49 years (83.3%) and had a disease duration shorter than five years (73.5%). We did not find any differences in terms of age and sex between the two groups of patients (*p* = ns). The main eosinophilic disease represented in the two populations was EoE (87.1%), followed by EGE (3.9%), and EC (8.9%). Finally, 55 (53.9%) patients were on proton pump inhibitor therapy, 22 (21.6%) followed a specific restrictive diet, 48 (47.1%) were taking topical steroid therapy and two (2.0%) patients were on biologic drugs (*n* = 1 Vedolizumab, *n* = 1 Omalizumab). Patients from the Campania region were more frequently following a diet, whereas patients from the Veneto region were more commonly undergoing medical therapy (*p* < 0.001).

According to our survey and telemedicine visits, no patient was diagnosed with COVID-19 infection through nasopharyngeal swabs, although seven patients had respiratory symptoms compatible with it. Moreover, four family members had respiratory symptoms suggestive of COVID-19 without firm confirmation of the diagnosis because of the poor symptomatology (Table 2). In addition, no EGID-related symptom recurrence has been recorded in our patients on maintenance therapy, with a rate of adherence to treatment of about 100%.

Table 2 reports the answers to the questions regarding COVID-19 infection and the related psychological perception in the overall population. As shown in Figure 2, most of our patients were worried because of the COVID-19 outbreak (question 1: enough 43.1%, much/very much 43.2%), social distancing, and movement to crowded places, such as supermarket or food shops (question 3: enough 32.4%, much/very much 51.9%), and felt disturbed and anxious because of the infection (question 5: enough 40.2%, much/very much 39.2%). However, most of our patients did not consider themselves at higher risk of contracting COVID-19 infection or any other kind of infection due to their chronic condition (question 2 and 7) or therapy (question 10: No 47%). Moreover, the majority of our patients were worried to go to hospital (question 9: No 61.8%), whereas they did not believe that the information provided by social and mass media about COVID-19 diffusion and complications were excessive (question 4: not at all 31.4%, little 37.2%). Finally, about 30% of patients perceived of not being well followed-up by physicians, but 87.2% agreed with telemedicine remote visits.

When we tested for differences in survey’s answers among subgroups we found the following results: women were more worried to be at risk due to their chronic disease (question 2: enough 44.1%, much/very much 38.23% in women vs. enough 32.35%, much/very much 17.64% in men, *p* = 0.024; Figure 3). Similarly, women felt to be more susceptible to infections (question 7, *p* = 0.02; Figure 3). Regarding social and mass media information patients from the Campania region (i.e., the area without lockdown) thought that the COVID-19 programs of information were more excessive than those from the Veneto region (question 4: enough 18.3% and much/very much 7.3% in Veneto patients vs. enough 35% and much/very much 20% in Campania patients, *p* = 0.02). More patients from the Veneto region (65%) were reluctant to go to hospital compared to the Campania ones (45%; *p* = 0.004). Finally, as shown in Figure 4, patients with a lower degree of education thought that they were at higher risk for their disease than those with higher level of education (*p* = 0.03) and, likewise, they felt more disturbed or worried about coronavirus infection (*p* = 0.04; Figure 4). Finally, we observed a low adherence to the seasonal flu vaccination (only 17.6% of patients) and only 46.1% of patients would undergo a coronavirus vaccination, when it will be available, while 47% were in doubt about it (Table 2).

## 4. Discussion

Eosinophilic gastrointestinal diseases are characterized by dense infiltration of eosinophils in gastrointestinal tissues, resulting in morphological and functional abnormalities of the gastrointestinal tract [18,19]. They are considered to be caused by chronic allergic reactions to various allergens, including food and environmental antigens. The pathogenesis of EoE has yet to be completely elucidated, with both genetic and environmental factors considered to be involved [20,21,22]. In case of EoE, medical therapies consist primarily of oral proton pump inhibitors, elimination diet or orally administered topical corticosteroids, whereas immunosuppressive or immunomodulatory drugs, such as systemic corticosteroids and biologic therapies, are more frequently prescribed in patients with EGE or EC [6,7,8,19,23,24,25,26]. Overall, these treatments, including proton pump inhibitors, have been associated with an increased risk of infection, particularly at respiratory and digestive tracts, and in turn to potential disease relapse (i.e., EGID) [27,28]. Thus, we decided to investigate the clinical impact of COVID-19 infection among our EGID patients, observing that after more than 40 days of COVID-19 outbreak, no cases of confirmed COVID-19 were found and no EGID-related symptom recurrence was recorded. As of 4 April, 2020, seven patients had respiratory symptoms compatible with coronavirus infection, but because of their mild entity and the lack of complications, together with the need of saving our resources for more severe cases, they did not undergo a confirmatory test (i.e., nasopharyngeal swab and/or serological test). There are various explanations for these results. First, from the beginning of the outbreak our patients were strongly invited to follow the governmental measures suggested for the prevention of SARS-CoV-2 infection, including quarantine and other personal measures (i.e., handwashing, sanitization, avoiding public and overcrowded places, and wearing surgical masks in every public place). In this regard, a recent rapid review from a Cochrane database, including 29 studies, 10 modelling studies on COVID-19, four observational studies, and 15 modelling studies on severe acute respiratory syndrome and Middle East respiratory syndrome, observed that when quarantine is associated with other prevention and control measures, including school closure, travel restrictions, and social distancing, a large effect on the reduction of new cases, transmissions, and deaths can be observed [29]. Second, a service of telemedicine was organized for direct communication or consultation to provide an effective triage, screening, and treatment method. Moreover, telemedicine reduced the number of in-person visits thereby lessening face-to-face contact among patients and physicians, further reducing the chance of infection transmission. Findings from different studies suggested that telemedicine and virtual care could be used by physicians to provide needed medical care to patients during the pandemic and beyond [30], and this was demonstrated also during COVID-19 pandemic [31]. Third, we recommended that each EGID patient had to continue their ongoing treatment regimens and obtained an extraordinarily high adherence (100%), thus reducing the risk of disease relapse. Finally, it is likely that the relatively young age of our population helped to minimize the risk of infection and complications, since the older age seems to be a key factor for increasing COVID-19-related morbidity and mortality [11,12].

Patients with EGIDs usually report impaired quality of life, due to both their disease and EGID-related therapies [32,33,34]. Thus, we have investigated the psychological impact of COVID-19 outbreak in our population. For doing this, we decided to include patients from two areas of Italy with different prevalence of COVID-19 infection, particularly to assess the effects of lockdown. After declaring SARS-CoV-2 a pandemic in March 2020, extraordinary measures to contain the viral spread, such as the lockdown of vast areas in China first and Korea and Italy afterwards were adopted, with the logical consequence of increasing the perception of the disease and its fear. These measures were also emphasized by the public services and mass media, reporting the daily raising rate of morbidity and mortality, especially in Italy, which probably generated further misconceptions and concerns. However, despite a general fear about the infection, the need of social distancing, and the denial of moving to crowded places, our patients with EGIDs did not consider themselves at greater risk of COVID-19 infection compared to the general population or felt to be more susceptible to any other kind of infection. Moreover, they did not consider their current therapy as a risk factor for increased morbidity. These data were reported by patients from both the Campania and Veneto regions, suggesting that the lockdown had a limited impact on disease perception by our patients. As further confirmation of this, about half of our patients declared that would not undergo a coronavirus vaccination, when available. Based on our findings, we believe that the efforts aimed to reduce the psychological burden [35] of this dramatic infection should be addressed more on the general population than on specific groups of patients with chronic disorders, although the latter continue to take potentially dangerous medication such as the immunosuppressive ones.

When we analyzed the differences among subgroups of responders to our survey, we found that women felt themselves to be at higher risk of any kind of infection, including the COVID-19 one, due to their chronic condition. Similar results have been observed in a recent survey of heath care workers in hospitals equipped with fever clinics or wards for patients with COVID-19 in Wuhan and other regions in China, where participants reporting greater psychological burden were more frequently women [35]. This is agreement with previous medical literature showing that female gender is more affected by anxiety and depression [36,37]. Moreover, as outlined by Lai et al. in their recent study [35], another important factor associated with increased distress due to COVID-19 infection is the educational level, since we observed that patients with lower degree of education felt to be more disturbed and anxious and thought to be at greater risk as compared to patients with higher degree of education. These results confirm previous findings emphasizing the protective effect of higher educational level against anxiety, depression and health status perception [38,39]. In summary, based on our results, female gender and low education level have a greater impact on COVID-19 perception compared to the lockdown status or other factors, like ongoing therapies and age, which were initially suspected to be more relevant.

We observed that only one factor distinguished EGID patients from the Campania and Veneto regions, in that people from Campania complained of the excessive social and mass media information about morbidity and mortality of COVID-19 infection and were not worried of going to hospital for scheduled diagnostic procedures or visits. This is probably due to the different rates of infection and mortality between the two Italian regions and, indeed, people from Veneto were reluctant to go to hospital in order to continue their regular follow-up. To note, about 30% of our patients were worried because they perceived the possibility of not being well followed-up during the emergency, independently of their region, but almost all were happy because we quickly arranged remote telemedicine consultations.

To date, data regarding the impact of Coronaviridiae infection in EGID patients taking chronic therapies for their conditions are lacking, and recommendations about their management during this pandemic are usually translated from those provided by inflammatory bowel disease patients, who share with EGIDs the inflammatory pathogenesis and treatment modalities. As previously mentioned, inflammatory bowel diseases have been recently associated to SARS-CoV-2 infection and its complications, hypothesizing that their therapies, mainly steroids and biologics, could promote SARS-CoV-2 progression and mortality [14,15]. Similarly, in EGID patients, previous studies associated steroid therapies with a potential increased risk of infection. For instance, Liacouras et al. observed that, despite the efficacy of these drugs in inducing clinical and histological remission in children with EGIDs, they were affected by relevant systemic adverse events [23]. In addition, the albeit small number of clinical studies regarding long-term topical corticosteroid use in EGIDs raised concerns regarding long-term adverse events, such as oral and esophageal candidiasis (up to 15% of cases) [40,41,42]. We also know from studies carried out in the general population or in patients with other immune-mediated diseases that corticosteroids or biologic therapies have been associated with an increased risk of infection (including flu) compared with those not taking these drugs [27,40,43,44,45]. Finally, Lukin et al. evaluating clinical outcomes between COVID-19 patients with and without inflammatory bowel disease observed that corticosteroids, and not biologics, were the drugs associated with higher morbidity among COVID-19 patients [15]. Despite the above finding, we did not observe any influence of these treatments on COVID-19 infection and disease outcome. It is noteworthy, however, that only a minority of our patients were treated with systemic steroids or biologic drugs. Moreover, it is relevant to emphasize that the opinions on the negative effects of the above therapies are not univocal with recent uncontrolled observations suggesting that immunomodulation could be even helpful in a subgroup of severe COVID-19 with hyper-inflammatory features [46]. Indeed, some current therapeutic efforts against COVID-19 infection are addressed towards the use of biologic drugs, such as those approved for the treatment of rheumatoid arthritis [47]. Moreover, previous experience with high-dose steroids in the treatment of SARS-CoV or Middle East Respiratory Syndrome (MERS)-CoV did not seem to worsen the clinical course of these diseases, although these drugs did not permit to clear the virus rapidly [48].

Furthermore, proton pump inhibitors have been associated with increased risk of gastrointestinal and pulmonary infections, although the data suggesting such association are conflicting [28,49,50,51]. As to pulmonary infections, various studies suggest a possible association between proton pump inhibitor use and development of community-acquired pneumonia, due to the potential ascending bacterial colonization from the gut as effect of the high intragastric pH induced by proton pump inhibitors and its invasion of upper airways. The first reports showed an increased risk of community-acquired pneumonia in outpatients taking proton pump inhibitors [52,53] and a subsequent meta-analysis of eight observational studies found that the overall risk of pneumonia was increased by 27% with proton pump inhibitor use [54]. In contrast, two meta-analysis of eight cohort studies and 10 randomized controlled trials, respectively, revealed that the acid suppressive drugs were not associated with increased risk of hospitalization for community-acquired pneumonia [55,56]. Likewise, the chronic proton pump inhibitor use has been also linked to small intestinal bacterial overgrowth, alterations of the intestinal microbiome and Clostridium difficile (CDI) infection [57,58,59]. In particular, the relative odds ratios for CDI reported in different meta-analyses are frequently greater than 2, and therefore this adverse event seems to be more consistent than the other harms reported for proton pump inhibitor therapy [28,49,50,60,61]. Overall, it can be hypothesized that proton pump inhibitor use could represent a risk factor for COVID-19 infection, particularly for patients with EoE, who mostly take these drugs. Moreover, it is relevant to note that recent data suggested that the viral SARS-CoV-2 RNA remains positive in feces in more than 20% of positive patients, even after negative conversion of viral RNA in the respiratory tract, thus indicating that the gastrointestinal viral infection may represent a transmission route of infection and migration into the respiratory tract [62]. Despite these data, we did not observe any influence of proton pump inhibitors on COVID-19 infection probability and disease outcome in our EGID patients, narrowing the magnitude of the association between proton pump inhibitor use and infections.

Our web-based survey highlights another problem related to facultative disease vaccination in Italy, like the case of flu vaccination. In Italy, doctors offer a flu shot to eligible patients at any time during the flu season. In particular, according to the Italian Ministry of Health subjects older than 65 years, children over six months, health care professionals and all subjects with a chronic disease, including patients with immune-mediated gastrointestinal disorders, may have a free flu shot. However, less than 20% of the general population undergo flu vaccination [63]. Thus, the answer to our question related to the willing of undergoing coronavirus vaccination when available is in line with what we observed for seasonal flu in our population. The reason of the low adherence is unclear, but it highlights the need for major efforts from the Italian Health Institutions to explain and emphasize the clinical benefit of disease vaccination. 

A limitation of our study is represented by the small number of patients suffering from eosinophilic gastroenteritis and eosinophilic colitis. However, these two entities are quite rare and predominate in children, while our centers are more dedicated to the cure of adult patients. Another limitation is the lack of objective assessment of COVID-19 infection among our patients through nasopharyngeal swabs or blood tests. However, this was not possible, because, at the time of our survey, all the regional health resources were allocated to the diagnosis and treatment of more severe patients with suspected or confirmed COVID-19 infection. Third, we did not use validated questionnaire for the evaluation of psychological burden. However, this was done in order to render our survey as rapid and easy to understand as possible in the attempt to minimize the number of non-responding patients. Indeed, a large number of patients accepted to answer our questionnaire, and this contributed to reduce the risk that only patients with more anxiety would have provided their feedback. To note, we observed that patients from the Campania region provided more responses compared to patients from the Veneto region, however, without reaching statistical significance. This is likely due to the different samples of the two cohorts, which allowed the investigators from the University of Salerno to contact more frequently and more efficiently the non-responder patients and to convince them to fill in the questionnaire.

Our study has several strengths. First, to the best of our knowledge, it is the first investigation to deal with the impact of COVID-19 infection on patients with EGIDs. Second, we investigated also the psychological impact of such infection among our patients with chronic gastrointestinal disorders on maintenance treatment. In particular, we included participants from two regions with quite different situations in terms of disease prevalence and lockdown order to evaluate better the influence of the different burden of infection and the restrictive measures adopted. Lastly, we performed the study as soon as the lockdown occurred, and when there was no information about its restraint times.

According to the results of our study, we conclude that COVID-19 infection has limited clinical and psychological impact on patients with EGIDs. Patients continued to take their usual therapies and we found that no patient contracted the SARS-CoV-2 infection, at least after more than 40 days of the COVID-19 outbreak. From the psychological point of view, our patients with EGIDs did not consider themselves at greater risk of COVID-19 infection compared to the general population or felt to be more susceptible to any other kind of infection. Moreover, we found that the lower degree of educational level and the female gender, but not the fact of living in a lockdown area, influenced the perception of SARS-CoV-2 infection in Italy. Further studies, including larger cohorts of patients and longer follow-up time periods, are mandatory to further elucidate the impact of COVID-19 infection on EGID patients.

## Figures and Tables

**Figure 1 jcm-09-02011-f001:**
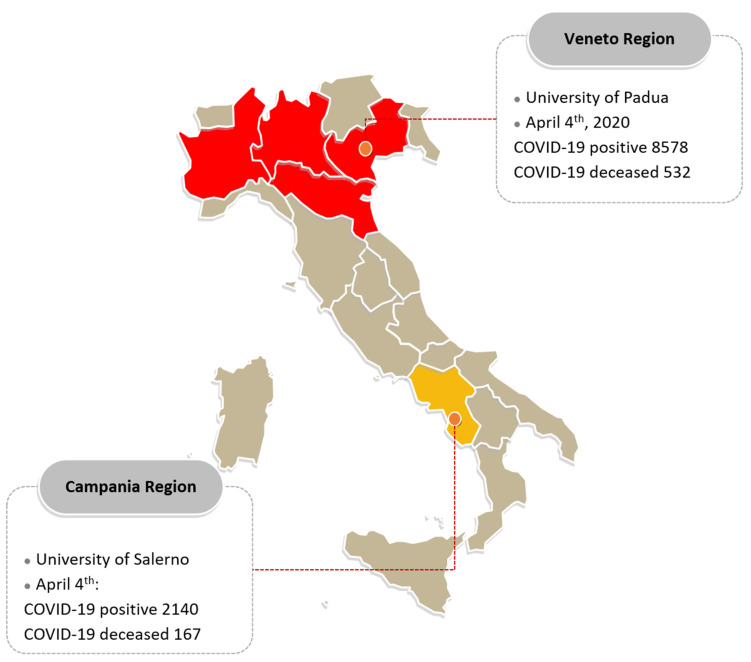
Italy map with COVID-19 spread updated on 4 April 2020 in Italian regions of Veneto and Campania.

**Figure 2 jcm-09-02011-f002:**
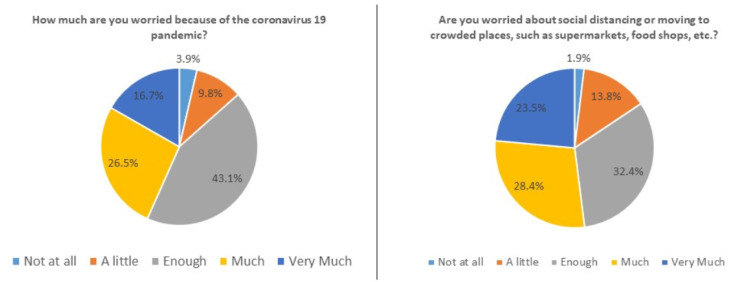
Perception of COVID-19 infection from patients with eosinophilic gastrointestinal diseases (EGIDs) in the Campania and Veneto regions of Italy.

**Figure 3 jcm-09-02011-f003:**
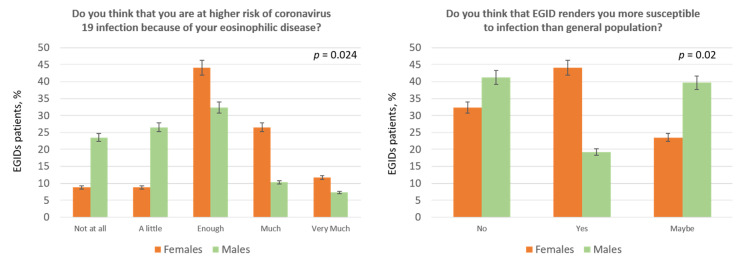
Perception of COVID-19 infection based on gender.

**Figure 4 jcm-09-02011-f004:**
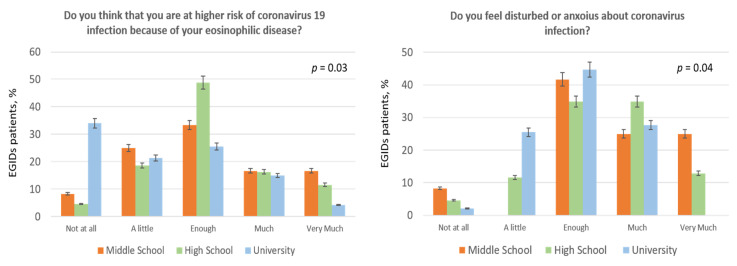
Perception of COVID-19 infection based on educational level.

**Table 1 jcm-09-02011-t001:** Demographic and clinical characteristics of eosinophilic gastrointestinal disease (EGID) patients stratified per each Italian region.

	Whole Population	Veneto Region	Campania Region	*p*-Value
No. of questionnaires received /sent (%)	102/157 (64.9%)	82/132 (62.1%)	20/25 (80.0%)	0.11
Female/Male, *n* (%)	34/68	30/52	4/16	0.15
	(33.3%/66.7%)	(36.6%/63.4%)	(20.0%/80%)	
Age groups, mean (SD)	36.6 (12.4)	35.5 (11.8)	40.9 (14.1)	0.08
- <35 years, *n* (%)	51 (50.0)	43 (52.3)	8 (40.0)	0.05
- 35–49 years, *n* (%)	34 (33.3)	29 (35.4)	5 (20.0)	
- ≥50 years, *n* (%)	17 (16.7)	10 (12.2)	7 (35.0)	
Education degree, *n* (%)				
- Middle school	12 (11.8)	10 (12.2)	2 (10)	0.18
- High school	43 (42.1)	31 (37.8)	11 (60)	
- University	47 (46.1)	41 (50)	6 (30)	
Rural area, *n* (%)	31 (34.4)	27 (32.9)	4 (20)	0.26
Urban area, *n* (%)	71 (69.6)	55 (67.1)	16 (80)	
EGID phenotype, *n* (%)				
- Eosinophilic esophagitis	89 (87.1)	70 (85.4)	19 (94.8)	0.31
- Eosinophilic colitis	9 (8.9)	9 (11.0)	0 (0.0)	
- Eosinophilic gastroenteritis	4 (4)	3 (3.6)	1 (5.2)	
Time from diagnosis,				
- <5 years, *n* (%)	75 (73.5)	57 (69.5)	18 (90)	0.15
- 5–9 years, *n* (%)	19 (18.6)	17 (20.7)	2 (10.0)	
- ≥10 years, *n* (%)	8 (7.4)	8 (9.8)	0 (0.0)	
Current therapy, *n* (%)				
- Diet	22 (21.6)	10 (12.2)	12 (60.0)	<0.001
- Proton pump inhibitors	55 (53.9)	49 (59.7)	6 (30.0)	
- Oral topical steroids	48 (47.1)	48 (58.5)	0 (0.0)	
- Systemic steroids (prednisone, budesonide)	10 (9.8)	9 (11.0)	1 (5.0)	
- Biologic drugs (vedolizumab, omalizumab)	2 (2.0)	2 (2.4)	0 (0.0)	
- Other (immunosuppressive drugs, sodium cromoglycate, anti-leukotrienes)	2 (2.0)	1 (1.2)	1 (5.0)	

SD = standard deviation.

**Table 2 jcm-09-02011-t002:** Survey administered to eosinophilc gastrointestinal diseases (EGIDs) patients for evaluating COVID-19 perception and infection occurrence.

	Not at all	Little	Enough	Much	Very Much
1. How much are you worried because of the coronavirus 19 pandemic?	3.9%	9.8%	43.1%	26.5%	16.7%
2. Do you think that you are at higher risk of coronavirus 19 infection because of your eosinophilic disease?	18.6%	20.6%	36.3%	15.7%	8.8%
3. Are you worried about social distancing or moving to crowded places, such as supermarkets, food shops, hospitals?	1.9%	13.8%	32.4%	28.4%	23.5%
4. Do you think that coronavirus 19 information from social and mass media are excessive?	31.4%	37.2%	21.6%	8.8%	1.0%
5. Do you feel disturbed or anxious about coronavirus infection?	3.9%	16.7%	40.2%	30.4%	8.8%
	I would like to speak with doctors	No, I do not feel myself well followed-up	Yes, it is perfect	I am afraid. I cannot tell everything	
6. Do you agree with telemedicine consultations?	8.8%	1.9%	87.4%	1.9%	
	No		Yes		Maybe
7. Do you think that EGID renders you more susceptible to infection than general population?	38.3%		27.4%		34.3%
8. Are you afraid that pandemic reduces your care by physicians to less than it would be necessary?	50.0%		33.3%		16.7%
9. Are you reluctant to go to hospital, because of coronavirus infection?	25.5%		61.8%		12.7%
10. Do you think to be more susceptible to infection than general population because of your current therapy?	47.0%		11.8%		41.2%
11. Did you undergo seasonal flu vaccination?	82.4%		17.6%		N/A
12. Would you like to undergo a coronavirus vaccination, when it will be available?	6.9%		46.1%		47.0%
13. Have you been diagnosed with COVID-19 infection (i.e., using nasopharyngeal swab)?	93.1%		0.0%		6.9% *
14. Have you been in contact with someone diagnosed with COVID-19 infection (i.e., using nasopharyngeal swab)?	96.1%		0.0%		3.9% *

* Patients reported flu-like symptoms or contact with persons presenting flu-like symptoms without further confirmation by nasopharyngeal swab. N/A= not applicable.

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
