# Peer review of "Clinical and Psychological Impact of COVID-19 Infection in Adult Patients with Eosinophilic Gastrointestinal Disorders during the SARS-CoV-2 Outbreak"

_jcm, 2020, doi:10.3390/jcm9062011_

Round 1

Reviewer 1 Report

Dr. Savarino and colleagues performed a prospective, observational, web-based study with EGID patients in order to assess the clinical impact of COVID-19 infection and the psychological perception of the COVID-19 emergency in this population of patients. The topic is current, the manuscript is well written. 

I have some minor comments:

  • were all the EGID patients at each study invited? Could the author comment on the difference between centers in patient participation (80 vs 62)?
  • Figure 2 depicts the perception of COVID-19 infection in the overall population; it could be interesting to assess if there were differences between the two centers and perhaps comment on this;
  • Table 2: I'd modify or remove the stars from the table as the rendering is not optimal;
  • discussion, row 4: it should be "more frequently prescribed";
  • could the authors explain and comment on the fact that about half of the patients would not undergo a coronavirus vaccination if it was available?
  • there are some typos that need to be fixed

Author Response

We thank this reviewer for his/her positive opinion on our paper.

C1. were all the EGID patients at each study invited? Could the author comment on the difference between centers in patient participation (80 vs 62)?.

R1. Thanks for your comment. We contacted all the EGID patients with available contact details as of 18 March 2020. The difference between centers in terms of response was not significant and so far we did not comment about that. However, it is true that patients from Campania Region provided more responses compared to patients from Veneto Region. This is likely due to the different samples of the two cohorts, which allowed the investigators from the University of Salerno to contact more frequently and more efficiently (by phone)  non-responder patients and to convince them  to fill in the questionnaire

C2. Figure 2 depicts the perception of COVID-19 infection in the overall population; it could be interesting to assess if there were differences between the two centers and perhaps comment on this

R2. Thanks for your comment. We checked about differences between the two cohorts according to disease perception, but we did not find any significant difference.

C3. Table 2: I'd modify or remove the stars from the table as the rendering is not optimal

R3. Thanks for your comment. We totally agree and indeed we removed the stars from the Table.

C4. discussion, row 4: it should be "more frequently prescribed"

R4. Thanks for your comment. We modified the sentence as suggested (page 7, row 181)

C5. could the authors explain and comment on the fact that about half of the patients would not undergo a coronavirus vaccination if it was available?

R5. Thank you for your comment. Indeed, our web-based survey highlights another problem related to facultative disease vaccination in Italy, in this case flu vaccination. In Italy, doctors offer flu shot to eligible patients at any time during the flu season. In particular, according to the Italian Ministry of Health subjects older than 65 years, children over 6 months, health care professionals and all subjects with a chronic disease, including patients with immune-mediated gastrointestinal disorders, have a free flu shot. However, less than 20% of the general population undergo flu vaccination in Italy (Coperture della vaccinazione antinfluenzale in Italia – EpiCentro. [cited 2018; Available from: https://www.epicentro.iss.it/influenza/coperture-vaccinali). Thus, response of our question related to the willing of undergoing coronavirus vaccination when available is in line with what we observed for seasonal flu in our population. The reason of the low adherence is unclear, but it highlights the need for major efforts from the Italian Health Institutions to explain and emphasize the clinical benefit of disease vaccination.

C7. there are some typos that need to be fixed

R7. Thanks for your comment. We revised the manuscript accordingly

Reviewer 2 Report

In this original article Savarino et al evaluated the impact of COVID 19 outbreak in a population of eosinophilic gastrointestinal diseases (EGIDs) by a survey. They investigated psychological worries and adherence to therapy. Overall, this is a good paper investigating an often neglected field of gastroenterology.

My only relevant criticism is linked to the fact that Authors did not investigate the perception of COVID 19 infection risk according to the type of medication prescribed for EGIDs. Indeed, at least for IBD, corticosteroids seem to be the only drug that increases the risk of infection (see Lukin DJ et al, Gastroenterology 2020) and should be tapered or withdrawn when possible according to expert consensus (see Hanzel J et al, Gastroenterology 2020). Please discuss.

Minor comments:

1) “EGID” in the title is an acronym that is hard to understand. Please replace it with its full meaning

2) Table 1: the “other” variable should be split in order to evaluate separately steroids, immune-suppressive drugs, biologics, cromoglycate…

Author Response

We thank this reviewer for his/her positive opinion on our paper.

C1. My only relevant criticism is linked to the fact that Authors did not investigate the perception of COVID 19 infection risk according to the type of medication prescribed for EGIDs. Indeed, at least for IBD, corticosteroids seem to be the only drug that increases the risk of infection (see Lukin DJ et al, Gastroenterology 2020) and should be tapered or withdrawn when possible according to expert consensus (see Hanzel J et al, Gastroenterology 2020). Please discuss

R1. Thanks for your comment. Unfortunately, the number of patients taking therapies different from PPIs or topical steroids was too small to permit an in-depth analysis based on the various medications. Anyway, we have included the relevant information on the role of steroid therapy in our manuscript, as suggested

C2. “EGID” in the title is an acronym that is hard to understand. Please replace it with its full meaning

R2. Thank you for your comment. We modified the title, accordingly

C3. Table 1: the “other” variable should be split in order to evaluate separately steroids, immune-suppressive drugs, biologics, cromoglycate…

R3. Thanks for your comment. We modified the Table accordingly. We checked about the role of steroid therapy, but the number of patients taking drugs different from PPI and oral steroids was too small for allowing a valid comparison and drawing any firm conclusion.